# Saharan dust contribution to the Caribbean summertime boundary layer - A lidar study during SALTRACE

Silke Groß[1], Josef Gasteiger[2], Volker Freudenthaler[2], Thomas Müller[4], Daniel Sauer[1], Carlos Toledano[3], and Albert Ansmann[4]

[1]Deutsches Zentrum für Luft- und Raumfahrt, Institut für Physik der Atmosphäre, Münchner Str. 20, 82234 Oberpfaffenhofen, Germany
[2]Ludwig-Maximilians-Universität, Meteorologisches Institut, Theresienstr. 37, 80333 München, Germany
[3]Universitat de Valladolid, Valladolid, Spain.
[4]Leibniz-Institut für Troposphärenforschung (TROPOS), Permoserstr. 15, 04318 Leipzig, Germany

*Correspondence to:* S. Groß (silke.gross@dlr.de)

**Abstract.** Dual-wavelength lidar measurements with the small lidar system POLIS of the Ludwig-Maximilians-Universität München were performed during the SALTRACE experiment at Barbados in June and July 2013. Based on high-accuracy measurements of the linear depolarization ratio down to about 200 m above ground level, the dust volume fraction and the dust mass concentration within the convective marine boundary layer can be derived. Additional information from radiosonde launches at the ground-based measurement site provide independent information on the convective marine boundary layer height and the meteorological situation within the convective marine boundary layer. We investigate the lidar-derived optical properties, the lidar ratio and the particle linear depolarization ratio at 355 and 532 nm and find mean values of 0.04 (stdev 0.03) and 0.05 (stdev 0.04) at 355 and 532 nm, respectively, for the particle linear depolarization ratio, and $(26 \pm 5)$ sr for the lidar ratio at 355 and 532 nm. For the concentration of dust in the convective marine boundary layer we find that most values were between 20 and 50 $\mu g/m^3$. On most days the dust contribution to total aerosol volume was about 30–40%. Comparing the dust contribution to the column-integrated sun-photometer measurements we see a correlation between high dust contribution, high total aerosol optical depth and a low Angström exponent, and of low dust contribution with low total aerosol optical depth.

## 1 Introduction

Saharan dust is one of the primary components of the global aerosol load (Forster and et al., 2007; Haywood and Boucher, 2000) with an estimated annual emission of more than 1000 Mt (Duce et al., 1991). Saharan dust can be transported over several thousand kilometers (Goudie and Middleton, 2001; Liu et al., 2008), influencing the Earth's energy budget on its way (Tegen et al., 1997). Turbulent downward mixing of dust in the convective marine boundary layer (CMBL) over the tropical Atlantic is assumed to be an efficient dust removal process. To support modeling efforts to simulate dust long-range transport and removal processes accurately and to validate the model output, a high-quality vertically resolved characterization of the optical and microphysical particle properties in the convective marine boundary layer and in the main dust layer (Saharan Air

Layer, SAL) is essential. SAL properties found during SALTRACE have already been presented by Groß et al. (2015), here we present the properties for the CMBL. Further papers will be presented in the framework of this SALTRACE special issue that will also partly deal with the removal of dust by turbulent downward mixing (Rittmeister et al., in preparation; Marinou et al., in preparation).

Typically, the cloud-topped convective boundary layer in the trade-wind-dominated tropics consists of the so-called sub-cloud layer, which is identical with the convective marine boundary layer in the absence of cloud formation and usually reaches to 500-1000 m height, and the cloud layer (Siebert et al., 2013). Moist convection can lead to a considerable increase of the overall depth of the convectively active height range. Over Barbados, cloud top heights were frequently observed from 1500-2500 m. In the convectively active zone, the observed particle linear depolarization is usually low ($< 0.1$) and rapidly

increases to typical values $> 0.2$ in the base of the elevated mineral dust layer (Groß et al. 2015, Haarig et al., in preparation for this special issue). This strong increase at the top of the cloud-topped or cloud-less CMBL can be interpreted as a clear sign for an efficient downward mixing of dust at the interface between the CMBL and the elevated SAL as it indicates large amount of dust below the well-defined SAL. Dust trapped in the CMBL will then be comparably quickly transported down to the ocean or land surface and deposited.

As concluded from observations of long-range transported dust with the space-borne lidar system CALIOP (Liu et al., 2008), the CMBL depth is low close to western Africa in summer and increases with distance from Africa. This finding is in agreement with shipborne lidar observations in May 2013 (Kanitz et al., 2014). Frequent cloud formation in the marine boundary layer change the thermodynamic conditions in the range from about 700 to 1500-2000 m towards a less stable air stratification such that subsequent cloud formation is facilitated and clouds may reach higher altitudes.

In this work we present information on the CMBL and the dust contribution within the CMBL over Barbados in June and July 2013. This information is based on ground-based lidar measurements with the depolarization and Raman lidar system POLIS (Freudenthaler et al., 2015; Groß et al., 2015) of the Ludwig-Maximilians-Universität, München, performed in cooperation with the Deutsches Zentrum für Luft- und Raumfahrt (DLR), and on radiosonde measurements (launched typically twice a day) over the Barbados ground-based site performed by the Leibnitz-Insitut of tropospheric research (TROPOS), Leipzig.

The measurements were conducted in the framework of the Saharan Aerosol Long-range Transport and Cloud-interaction Experiment (SALTRACE, Weinzierl et al. 2016). The measurement site was located in the area of the Caribbean Institute of Meteorology and Hydrology (CIMH) at Husbands ($13.14°$ N, $59.62°$ W, $100\,\mathrm{m}$), on the south-western part of Barbados.

The general properties discussed in this paper include height, wind direction and wind speed within the boundary layer. The lidar derived properties include the mean lidar ratio and mean particle depolarization ratio within the CMBL at 355 nm and

532 nm, and the retrieved dust volume fraction and dust concentration within the CMBL. The aim of this paper is to provide a detailed characterization of the CMBL as part of the vertical aerosol distribution over Barbados during SALTRACE. The characterization of the whole atmospheric column by vertical profiles of its properties, and a characterization of the different layers within is important as it enables to link ground-based dust measurements, airborne measurements, and column-integrated measurements e.g. with sun-photometers.

This paper is structured as follows; in Section 2 measurements, instrumentation, and methodology are briefly described, in Sections 3 and Section 4 the results are presented and discussed, and Section 5 concludes this work.

## 2 Instruments and Method

### 2.1 SALTRACE

The presented study is based on measurements during the Saharan Aerosol Long-range Transport and Aerosol-Cloud-Interaction Experiment (SALTRACE). Measurements were performed at Barbados in June and July 2013. SALTRACE was designed as closure experiment combining airborne and ground-based in-situ and remote sensing measurements with long-term observations and modeling efforts. For a detailed description of the SALTRACE project see Weinzierl et al. (2016).

### 2.2 POLIS lidar system

The lidar measurements used in this study were performed with the small six-channel dual-wavelength depolarization and Raman lidar system POLIS which was developed and build by the Meteorological Institute of the Ludwig-Maximilians-Universität, München (Groß et al., 2015; Freudenthaler et al., 2015). POLIS measures simultaneously the co-and cross-polarized laser return at 355 and 532 nm, during night-time additional measurements of the $N_2$-Raman shifted returns at 387 and 607 nm are performed. POLIS has a full overlap at about 200–250 m (depending on system configuration) allowing
studies close to the lidar system (i.e. in the boundary layer). The raw data resolution of the POLIS measurements is 3.75 m along-side and typically 10 s in time. Night-time measurements were used to derive the extinction coefficient and the lidar ratio based on the Raman approach described by Ansmann et al. (1992). The typical resolution for the analysis of the Raman measurements is 1–2 h in time and about 200 m along-side. The retrieved lidar ratio was then used for a Fernald/Klett-analysis (Klett, 1985; Fernald, 1984) of the elastic channels to derive extinction coefficient, backscatter coefficient and depolarization
ratio with much better vertical and temporal resolution than possible from the inelastic Raman channels. The vertical and temporal resolution of the measurements used for this study are about 155 m and about 30 minutes. In single cases measurement periods of about 1 h are used. For this study we used, when available, one day-time and one night-time measurement per day, coordinated with radiosonde ascents at the lidar site. For a general description of measurements and analysis see Groß et al. (2015).

### 2.3 CIMEL

Two AERONET sunphotometer (site name: Barbados_SALTRACE) were co-located with the ground-based lidar POLIS. Direct Sun observations were performed every 3 minutes (high frequency acquisition mode of the Cimel sunphotometers). They provide aerosol optical depth at 8 spectral channels in the wavelength range 340-1640nm. The multi-angle and multi-spectral measurements of sky radiance (almucantar and principal plane geometries, measured once every hour) are used to derive a set
of optical and microphysical aerosol properties by means of inversion algorithms: volume particle size distribution, complex

refractive index, single scattering albedo and fraction of spherical particles. For further details on the instrument, calibration procedures and data products, see Holben et al. (1998), Dubovik and King (2000), and Dubovik et al. (2006).

## 2.4 Radiosondes

During SALTRACE radiosondes were launched typically twice a day along with the lidar measurements; one during the morning sessions (14–15 UTC) and one during the evening session (23–1 UTC). The radiosondes recorded temperature, air pressure, relative humidity, wind speed and wind direction. Altogether 56 radiosondes were launched during SALTRACE.

## 2.5 In-situ measurements

In-situ dust mass concentrations were determined at the field station Ragged point (see Prospero and Mayol-Bracero 2013). For the SALTRACE campaign the station was equipped with additional instrumentation for dust characterization.

Aerosols were sampled through an aerosol PM10 inlet on top of a mast 17 m above ground. The aerosol laden air was drawn with a flow rate of 16.6 l/min through a 3/4 inch stainless steel tube. At the base of the mast the flow was split and directed to different instruments. Before the measurements, the aerosol was dried to a relative humidity of about 40% using Nafion membranes. Instruments for investigating properties of cloud condensation nuclei are described in Kristensen et al. (2016). Instruments for measuring optical properties were placed in an outside cabinet under the mast to avoid sampling line losses for super-micrometer particles. Particle light scattering coefficients were measured with a Nephelometer (Aurora4000m, Ecotech Pty Ltd., Australia) and light absorption coefficients were measured with a Spectral Optical Absorption Photometer (SOAP, Müller et al. 2011). Additionally, an Aerodynamic Particle Sizer (APS-3321, TSI) for measuring the particle number size distribution for sub-micrometer particles and a Scanning Mobility Particle Sizer (SMPS, Wiedensohler et al. (2012)) for measuring the sub-micrometer particle number size distribution were used.

The total particle mass concentration is derived from the particle number size distribution measured with APS and SMPS. To convert the aerodynamic particle number size distribution to a volume equivalent number size distribution for dust particles we chose a dynamic shape factor of 1.17 and a density of 2.45 g/cm$^3$ . The method and the values for the dynamic shape factor and density are discussed in Niedermeier et al. (2014). For cases with high dust concentrations (dust mass concentration > 20 $\mu g/m^3$), the total mass concentration from the particle number size distribution was found to be lower by 14% compared to the dust mass concentration from optical absorption. Niedermeier et al. (2014) pointed out, that the determination of the dust mass concentration from optical absorption depends on the choice of the mass absorption coefficient, specifically on the relative abundance of the strongly absorption iron oxides. The mass absorption coefficient used in this study was derived from the SAMUM-I campaign for Saharan desert dust with an relative iron abundance of 1% (Kandler et al., 2009). This can explain differences between the two methods qualitatively, a detailed error discussion of the methods is beyond the scope of this paper.

## 2.6 Aerosol type separation

From the profiles of the particle linear depolarization ratio $\delta_p$ and particle backscatter coefficient $\beta_p$ measured at 532 nm we determine the dust backscatter coefficient $\beta_d$ following the procedure described by Tesche et al. (2009a); Groß et al. (2011a) and Ansmann et al. (2011). This method is based on the work of Shimizu et al. (2004) assuming a two component mixture with known particle depolarization ratio of the two components as indicated by coordinated aircraft in-situ measurements (Weinzierl et al., 2016). As input parameters for the aerosol type separation we use the mean dust linear depolarization ratio $\delta_d$ of 0.30 as found from the former field campaigns during the Saharan Mineral Dust Experiment (SAMUM) (Freudenthaler et al., 2009; Groß et al., 2011b). We also include measurements over Barbados during the SALTRACE experiment focusing on the general situation during the campaign and on the optical properties of long-range transported Saharan dust (Groß et al., 2015). For marine (non-dust) aerosols we use a mean linear depolarization ratio $\delta_{nd}$ of 0.02 for relative humidity values $\geq 45\%$ (Groß et al., 2011b, 2015). The dust $\alpha_d$ and marine extinction coefficient $\alpha_{nd}$ are calculated according to $\alpha = \beta \cdot S$ with a dust lidar ratio $S_d$ of 55 sr (Tesche et al., 2009b; Groß et al., 2015) and a marine lidar ratio $S_{nd}$ of 20 sr (Groß et al., 2015) at 532 nm.

## 2.7 Dust volume and mass conversion

In a next step the volume concentration of both particle types is derived using a conversion factor from extinction to volume concentration $v/\alpha$. This conversion factor strongly depends on the microphysical properties of the aerosol type, in particular on the size distribution. To derive $v/\alpha$ and to study the potential differences in the conversion factor $v/\alpha$ for fresh and long-range transported dust, $v/\alpha$ was derived from SAMUM and SALTRACE-AERONET measurements during pure dust periods. The conversion factors are derived from AERONET aerosol mixture retrieved by the AERONET inversion algorithm (Mamouri and Ansmann, 2016) which is then used to derive the volume and extinction of this aerosol mixture to finally derive $v/\alpha$. For the dust conversion factor well-defined dust outbreaks observed over Morocco, Cape Verde and Barbados with aerosol optical depth (AOD) > 0.2 at 500 nm and Angström Exponent (AE) < 0.2 (Barbados) and < 0.4 (Cape Verde and Morocco) are considered. The AERONET-derived conversion factors show a constant value of $v/\alpha = 0.65 \cdot 10^{-6}$ m for measurements close to the source and over Barbados. In this study we adopt this value for the conversion. To derive the conversion factor for marine aerosols AERONET measurements at Barbados from 2007–2015 were analyzed. Altogether about 100 observations which were defined as pure marine (AOD < 0.07 at 500 nm and AE between 0.25 and 0.6) were found, resulting in a marine conversion factor $v/\alpha = 0.66 \cdot 10^{-6}$ m. The conversion factors for dust and marine aerosols are similar because the size distribution of both aerosol types are rather similar. The dust volume fraction can then easily be calculated from the ratio of the dust volume concentration to the total volume concentration. The dust mass concentration is calculated from the dust volume concentration by multiplication with the particle density, which we assumed to be 2.5 g/cm$^3$ based on the fact, that in most cases a mixtures of coarse and fine mode particles (Ansmann et al., 2011; Mamouri and Ansmann, 2014) and sulfate particles (Kaaden et al., 2009) was found in the Saharan dust layer.

## 2.8  CMBL height identification

We infer the height of the CMBL from the radiosonde measurements. As turbulent convection in the boundary layer causes constant values of the potential temperature and mixing ratio (Hooper and Eloranta, 1986; Kaimal et al., 1976), we use these properties as indicators for the CMBL height. Furthermore we use temperature information derived from the radiosonde as-
cents, as the boundary layer is mostly capped by an inversion layer (Carson, 1973). Independently we derived the CMBL height from the optical properties measured by our lidar system. Aerosol concentration and thus the aerosol backscatter co-efficient show a gradient on top of the boundary layer (Boers et al., 1984; Hooper and Eloranta, 1986). We use the strongest gradient to determine the CMBL height. Furthermore we also use the measured intensive lidar quantities like the linear particle depolarization ratio to strengthen our result.

## 3  Results

### 3.1  General measurement situation

The main measurement period for closure studies during SALTRACE was from 20 June to 12 July 2013. During this time the aerosol situation above Barbados was characterized by an aerosol optical depth (AOD) which mainly ranged between about 0.2 and 0.4 almost wavelength-independent for the CIMEL measurements at 340, 500 and 1020 nm (Fig. 1). The Angström
Exponent (AE) was typically 0.2 or lower. On some days, however, the aerosol load over Barbados was substantially lower, with AOD values well below 0.2. Those low AOD values were connected with higher AE of about 0.5–0.7. On 7 July 2013 the AE was even as high as 1.1. Five episodes with high AOD values (up to 0.6) were observed.

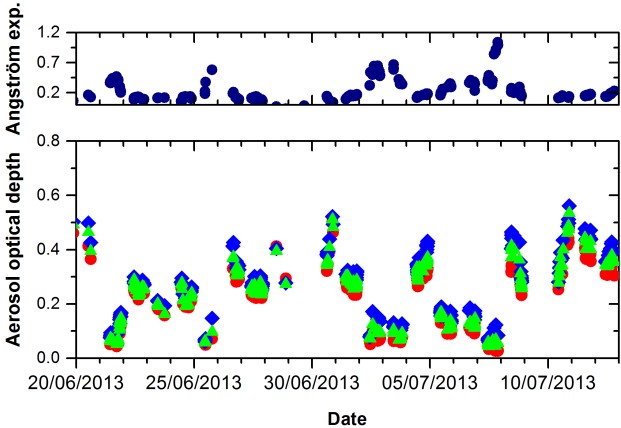

**Figure 1.** Angström Exponent between 440 and 870 nm (upper panel) and the Aerosol optical depth at 340 (blue), 500 (green) and 1020 nm (red) derived from AERONET-CIMEL sun-photometer measurements at the Caribbean Institute for Meteorology and Hydrology, Barbados (data: Barbados_SALTRACE) from 20 June to 13 July 2013

## 3.2 Case study – 10 July 2013

A multi-layer aerosol structure was observed over Barbados during SALTRACE in June and July 2013 (Groß et al., 2015). Figure 2 provides an overview of the situation on 10 July 2013 based on aerosol lidar observation at 532 nm with POLIS. The general structure of the shown profiles is representative for most of the SALTRACE measurements. From the volume and particle linear depolarization ratio (Fig. 2b) two aerosol regimes are clearly visible; in the lowermost 1.5 km the particle depolarization ratio is around or even below 0.1, indicating that dust has only a minor contribution to the aerosol mixture. Above about 1.6 km the volume and particle linear depolarization ratio is clearly higher with values of about 0.14 to 0.18 for the volume linear depolarization ratio and 0.28 to 0.3 for the particle linear depolarization ratio. The separation of the two aerosol regimes is also visible in the radiosonde measurements launched at 17:49 UTC (Fig. 2e,f) showing a temperature inversion between about 1.6 and 1.8 km height together with a change in relative humidity, mixing ratio, potential temperature and wind speed (not shown).

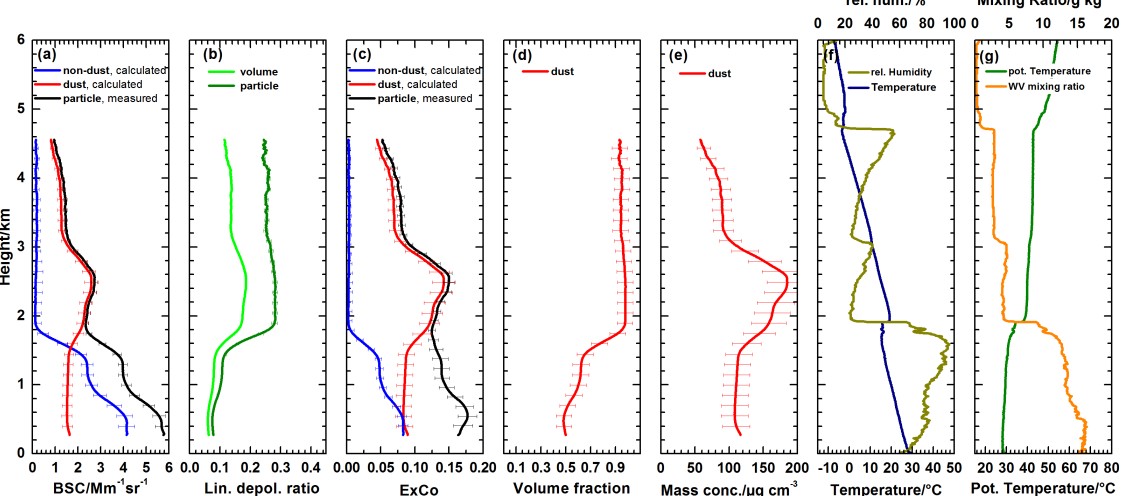

**Figure 2.** (a) Separation of dust (red) and non-dust (blue) particle backscatter coefficient (BSC) at 532 nm from total particle backscatter coefficent (black) at 532 nm, (b) measured volume (light green) and particle (dark green) linear depolarization ratio at 532 nm, (c) dust (red) and non-dust (blue) extinction coefficient at 532 nm and the total particle extinction coefficient (black) at 532 nm derived from POLIS lidar measurements, (d) dust volume fraction, (e) dust concentration derived from POLIS measurements at Barbados on 10 July 2013 at 20 UTC, (f) temperature (dark blue) and relative humidity (dark yellow) profiles, and (g) profiles of potential temperature (dark green) and water vapor mixing ratio (orange) derived from radiosonde measurements started on 10 July 2013 at 17:49 UTC. Error bars give the systematical uncertainties resulting from measurement uncertainties and uncertainties in the lidar specific input parameters.

According to the criteria for CMBL height identification with lidar (Boers et al., 1984; Hooper and Eloranta, 1986) the CMBL height on 10 July 2013 was at about 500 m. Lidar measurements show a strong gradient in the aerosol backscatter coefficient (Fig. 2a) at about 500 m. CMBL height detection from radiosonde measurements at 17:49 UTC is not that distinctive;

only the water vapor mixing ratio shows slight changes. The capping inversion in the temperature profile and the change in the potential temperature on top of the CMBL (at about 500 m altitude) were missing or not pronounced. Also the measured volume and particle linear depolarization ratio (Fig. 2b) within the CMBL showed only slight differences compared to the values found in the transition layer between the CMBL and the Saharan air layer. The CMBL was characterized by a constant value

5   of the potential temperature of about 28°C and a constant water vapor mixing ratio of about 16 g/kg. The relative humidity within the boundary layer increased with height from about 65 to 80%. The wind in the boundary layer (not shown) came from easterly directions with a mean wind speed around 7 m/s. The aerosol optical depth within the boundary layer is about 0.1 at 532 nm, and the mean volume and particle depolarization ratio at 532 nm are about 0.06 and 0.08, respectively.

Following the procedure described in Section 2.7 the volume fraction of dust is derived (Fig. 2d). Above about 1.6 km dust

10  contributes to almost 100% to the total aerosol volume while below 1.5 km height the dust contribution is only 70% or less. In the CMBL (lowermost 0.5 km) the dust volume fraction is even less than 60%. For the dust concentration (Fig. 2e) we find a rather constant value of about 110 $\mu$g/m$^3$ in the lowest 1.5 km. Above a height of 1.5 km the profile of the dust concentration generally follows the profile of the backscatter coefficient (Fig. 2a) and the extinction coefficient (Fig. 2c) with a maximum in dust concentration of about 190 $\mu$g/m$^3$ between 1.4 and 1.6 km height.

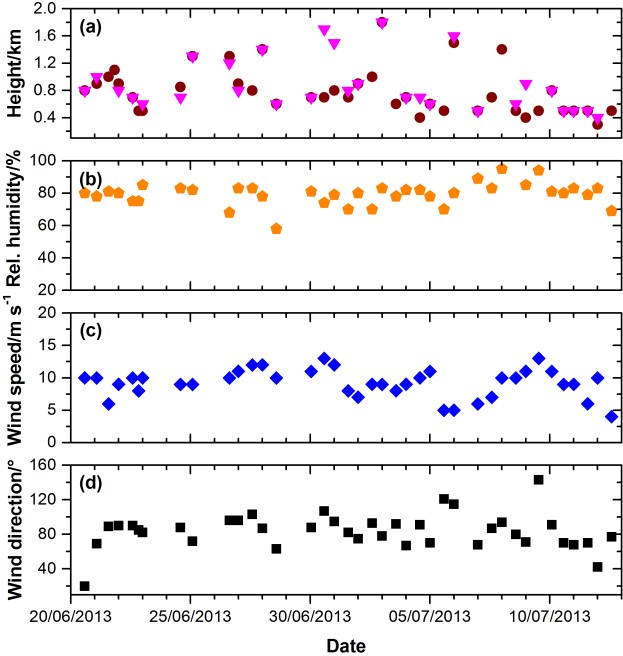

**Figure 3.** (a) CMBL height derived from radiosonde measurements (brown dots) and lidar measurements (magenta rectangles), (b) mean relative humidity within the CMBL derived from radiosonde measurements, (c) mean wind speed in the CMBL measured with radiosonde, and (d) mean wind direction in the CMBL derived from radiosonde measurements.

### 3.3 CMBL height and conditions during SALTRACE

As described in Section 2.8 the CMBL height is derived from radiosonde and lidar measurements (Fig. 3a). The CMBL top above our measurement site during SALTRACE is mainly between 0.5 and 0.9 km, in some cases the boundary layer reaches up to 1.8 km. The CMBL height derived from radiosondes and lidar mostly agree within 100 m, however in a few cases the CMBL height derived from lidar measurements is higher than the CMBL height derived from radiosondes. In those cases the CMBL height from radiosondes is mainly based on observed changes in the mixing ratio, we do not see a profound change in the potential temperature nor a capping inversion layer on top of the CMBL. However we do see a pronounced temperature inversion and change in the potential temperature in those cases within the height ranges of the lidar-derived CMBL heights. We frequently found pronounced changes of the intensive lidar quantities (i.e. quantities that only depend on the observed aerosol type or mixture but not on its amount) on top of the CMBL in connection with strong capping inversions and changes in potential temperature on top of the CMBL. The differences of the intensive optical properties between CMBL and transition layer was not as pronounced in situations where the capping inversion on top of the CMBL was absent and changes in potential temperature were small. Those later situations occurred mainly during the daytime.

During SALTRACE the relative humidity within the CMBL was quite high with values around 80%. Only on a few days the relative humidity within the boundary layer was lower, but never below 60% (Fig. 3b). The mean wind speed within the CMBL over Barbados during SALTRACE was predominantly between 8 and 12 m/s (Fig. 3c) which are optimal conditions for the production of sea salt particles (Gong, 2003; Knippertz et al., 2011). Between 5 July and 7 July 2013 and during individual days between 20 June and 23 June 2013 the wind speed was lower with values around 6 m/s, but those values are still above the empirically derived threshold for sea-salt aerosol production of 5.5 m/s (Knippertz et al., 2011). Only on the last days of our measurements the wind speed is lower than this empirical value.

From our measurements we assume that sea salt particles at Barbados exist only below about 1500 to 2000 m, most likely advected to our measurement site by strong onshore easterly winds. The wind direction within the CMBL derived from radiosondes (Fig. 3d) was mainly East to North-East, except for very few cases when the wind came from either northern or southern directions.

### 3.4 Optical properties within the CMBL

Aside from the last dust event during the SALTRACE campaign, the CMBL during SALTRACE was characterized by low values of the particle linear depolarization ratio and the lidar ratio ranging between 0.01–0.08 and 15–31 sr at 355 and 532 nm (Fig. 4). These values are in good agreement with the findings of Murayama et al. (1999), who found mean values of the particle linear depolarization ratio at 532 nm of 0.01 and 0.1 in a marine boundary layer without and with dust influence. They related the lower values to pure marine aerosols whereas they assumed a contribution of dust for the higher values. Similar to Murayama et al. (1999) and Groß et al. (2011b, a) we assign values of the particle linear depolarization ratio of 0.01–0.03 (mean value 0.02) at 355 and 532 nm to marine aerosols, which is in good agreement to the findings of Sakai et al. (2010), who found values of 0.01 for sea-salt droplets from laboratory chamber measurements at 532 nm. The corresponding values

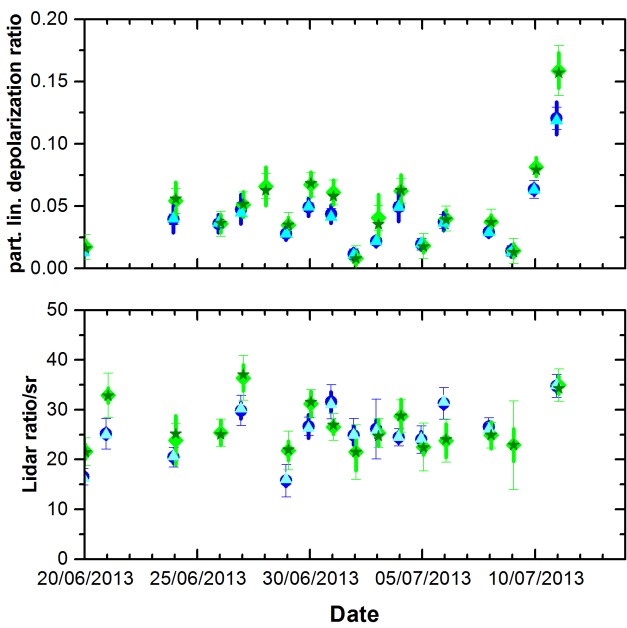

**Figure 4.** Mean values, median, standard deviation of the mean and mean systematic error of the particle linear depolarization ratio (upper panel) and the lidar ratio (lower panel) at 355 and 532 nm measured with POLIS within the convective marine boundary layer. Blue dots and green diamonds show the mean values at 355 and 532 nm, light blue triangles and dark green stars show the median values at 355 and 532 nm, thick lines show the standard deviation of the mean, and the thin error bars denote the mean systematic errors.

of the lidar ratio for the marine aerosol cases are 16–24 sr with mean values of $20 \pm 3$ at 355 nm and of $22 \pm 5$ at 532 nm. These values are also in good agreement to previous findings from theoretical studies (Ackermann, 1998) and measurements (Groß et al., 2011b, a). During the last measurement day the mean value of the lidar ratio within the CMBL is about 35 sr independent of wavelength. The particle linear depolarization ratio shows mean values of about 0.12 at 355 nm and of about

5    0.16 at 532 nm, indicating a larger fraction of mineral dust (Murayama et al., 1999; Groß et al., 2011b). The overall values of the particle linear depolarization ratio are wavelength independent with mean values of $0.04 \pm 0.03$ at 355 nm and $0.05 \pm 0.04$ at 532 nm ($\pm$-values give the standard deviation of the mean). The overall mean value of the lidar ratio within the boundary layer is $26 \pm 5$ sr at 355 and 532 nm ($\pm$-values give the standard deviation of the mean).

     According to former findings (e.g. Ackermann 1998; Sakai et al. 2010; Murayama et al. 1999; Groß et al. 2011b, a) and lidar

10    based aerosol classification schemes (Groß et al., 2011b; Burton et al., 2012; Groß et al., 2013) we find four days during the whole SALTRACE measurement period with a pure marine CMBL, one day with dust dominated dust-marine mixture within the CMBL. On all the other days we assign to polluted marine-dominated mixture within the CMBL. The results of the optical properties within the CMBL during SALTRACE are summarized in Table 1.

**Table 1.** Mean values of the lidar ratio and particle linear depolarization ratio (PLDR) including the mean systematic errors (±) for different aerosol types and their dominant time periods. *±-values for the overall mean values indicate the standard deviation of the mean.

| Dominant type | Date | PLDR | | Lidar ratio/sr | |
|---|---|---|---|---|---|
| | | 355 nm | 532 nm | 355 nm | 532 nm |
| marine | 20, 29 June 2, 5, 9 July | $0.02 \pm 0.01$ | $0.02 \pm 0.01$ | $20 \pm 3$ | $22 \pm 5$ |
| dust and marine (marine-dominated) | 24 June - 10 July without 2, 5, 9 July | $0.04 \pm 0.01$ | $0.05 \pm 0.01$ | $27 \pm 3$ | $28 \pm 3$ |
| dust and marine (dust-dominated) | 11 July | $0.12 \pm 0.01$ | $0.15 \pm 0.02$ | $35 \pm 3$ | $35 \pm 3$ |
| overall | | $0.04 \pm 0.03$ | $0.05 \pm 0.04$ | $26 \pm 5$ | $26 \pm 5$ |

**Figure 5.** Dust volume fraction (upper panel) and dust mass concentration (lower panel) within the CMBL over Barbados derived from POLIS lidar measurements. Error bars give the systematical uncertainties resulting from measurement uncertainties and uncertainties in the lidar specific input parameters.

### 3.5 Dust contribution in the CMBL

Figure 5 shows the mean dust volume fraction and the mean dust mass concentration within the CMBL retrieved from our lidar measurements. The dust volume fraction within the CMBL shows values between 0.01 and 0.65. For the majority of days we find values between about 0.3 and about 0.4. The dust mass concentration within the CMBL ranges between 2 and 100 $\mu$g/$m^3$ with most frequent values found between 20 and 50 $\mu$g/$m^3$. High dust mass concentrations are derived at the end of the campaign when the wind speed in the boundary layer was low. Thus we conclude that low wind in the CMBL may provide optimal conditions for dust downward mixing. Comparing the dust volume fraction and dust mass concentration within the CMBL with total AOD at 500 nm and AE (between 440 and 870 nm) for the complete atmospheric column derived from sun-photometer measurements (Fig. 1), one can see that days with low dust volume fraction and low dust mass concentration within the CMBL are found for days with column integrated AOD $\leq 0.1$ at 500 nm and corresponding AE $\geq 0.4$. The highest values of the dust volume fraction and the dust mass concentration are found for days with high AOD $\geq 0.4$ at 500 nm and corresponding low AE of $\leq 0.2$.

## 3.6   Closure

To validate the lidar derived dust contribution (i.e. dust volume fraction and dust mass concentration), we compared the lidar derived dust mass concentration with synchronized ground-based in-situ measurements of dust concentration at Ragged Point (Kristensen et al. 2016; Müller et al., in preparation for this special issue) at the eastern coast of Barbados (see Fig. 6).

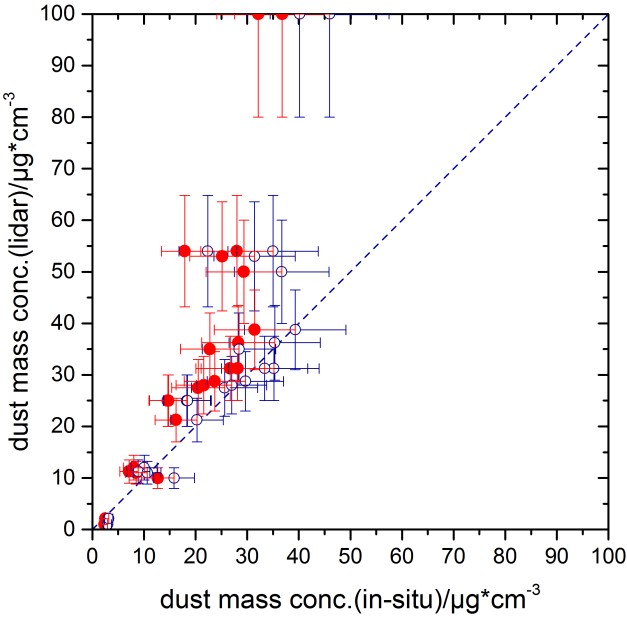

**Figure 6.** Dust mass concentration derived from lidar analysis versus dust mass concentration measured in-situ at Ragged Point (red dots). Only measurements that match in time (i.e. the lidar measurements were performed within or with temporal overlap to the sampling period of the in-situ measurements) are plotted. Additionally the in-situ vs. lidar derived dust mass concentration is plotted including a correction of the in-situ measurements (multiplied by a factor of 1.25) due to the nominal cut-off of the PM10 measurements (blue circles). Error bars for the lidar derived dust mass concentration give the systematic uncertainties resulting from measurement uncertainties and lidar specific input parameters. Error bars for the in-situ derived dust mass concentration result from the nominal cut-off and the assumed dust mass which thus can not be covered by the in-situ measurements. The dashed line gives the one-to-one line.

The dust concentration derived from both methods show good agreement for dust mass concentrations below about 40 $\mu g/m^3$. Dust mass concentrations derived from both methods below this value do not show significant differences. However, the lidar-derived dust concentration is on average about 3 $\mu g/m^3$ higher than the in-situ derived dust mass concentration. A linear fit considering all measurements with dust mass concentration below 40 $\mu g/m^3$ (including 17 measurement points) has a slope of 0.93. The good agreement of the derived dust mass concentration from lidar and in-situ measurements suggests that the method to derive dust mass concentration from lidar measurements works well.

For values above 40 $\mu g/m^3$ we do not see a good agreement between both methods. These larger dust mass concentrations derived from the lidar measurements at the western part of the island are not captured by the in-situ measurements. We found

evidence that the PM10 inlet causes an underestimation of mass concentrations especially for high dust concentrations. A detailed analysis for SALTRACE data is not finished yet, but unpublished data indicate that mass concentrations could be underestimated by up to 50%. Corrections are considered to be uncertain since inlet efficiencies are not well characterized for the high wind speeds frequently found at Ragged Point. This underestimation would partly explain the differences found
for mass concentrations above 40 $\mu g/m^3$. Furthermore, the ambient aerosol was sampled through a PM10 inlet (i.e. nominal cut-off at r=5$\mu$m) for the in-situ measurements (Kristensen et al., 2016). Thus a certain fraction of the ambient dust mass is likely not covered by the in-situ measurements. Using the OPAC desert mixture and assuming that particles up to r=10$\mu$m reach the Barbados measurement site (Weinzierl et al., 2016), we deduce that the mass concentration derived from the in-situ measurements needs to be multiplied by a factor of about 1.25 to get ambient mass concentration. However, as the coarse mode
size distribution of transported dust is not well characterized, we assume an uncertainty of $\pm 0.25$ for that factor. Taking this factor into account we not longer see a bias between the in-situ and lidar-derived dust mass concentration found for dust mass concentration higher than 40 $\mu g/m^3$.

## 4 Discussion

To characterize the optical properties of the convective marine boundary layer and the contribution of dust within the CMBL a
15 number of input parameters had to be adopted to derive dust volume and dust mass concentration within the CMBL.

For the separation of the different aerosol types contributing to the aerosol mixture in the CMBL we assumed a two-component mixture. Comparisons with coordinated aircraft in-situ measurements of the microphysical and the chemical properties of the observed aerosols justify the assumption of a two-component mixture of marine aerosols and mineral dust for the period of long-range dust transport over the Atlantic Ocean during summertime (Weinzierl et al., 2016), as it was the case in
this study. To derive the dust and marine backscatter and extinction coefficient the linear depolarization ratio and lidar ratio of dust and marine aerosols is needed. Those properties depend on the chemical and microphysical particle properties such as particle size and shape. These properties may change during long-range transport. The particle linear depolarization ratio as well as the lidar ratio of Saharan dust was studied during several field experiments. Freudenthaler et al. (2009) found a mean linear particle depolarization ratio of 0.31 $\pm$ 0.01 for fresh Saharan dust close to the source. This value does not change signif-
icantly for Saharan dust at the beginning of its long-range transport across (Groß et al., 2011a). Measurements of long-range transported Saharan dust over Europe (Wiegner et al., 2011), the Mid-West and the Caribbean (Burton et al., 2015) confirm a particle linear depolarization ratio of about 0.3 at 532 nm considering the systematic uncertainties of the lidar systems. Furthermore these optical properties of long-range transported Saharan dust were also seen with lidar measurements at Barbados during SALTRACE (Groß et al., 2015). The particle linear depolarization ratio ranged from 0.26 and 0.3 at 532 nm. Thus the
adopted value of 0.3 $\pm$ 0.01 in this study is in good agreement to previous findings. The lidar ratio of long-range transported Saharan dust is also derived from SALTRACE measurements. A mean value of 55sr at 532 nm was found (Groß et al., 2015) which is also in good agreement to previous studies of Saharan dust lidar ratios (e.g. Tesche et al. (2011)). In this study we use a value of 55 sr to derive the dust extinction coefficient. Denjean et al. (2015) show that long-range transported dust does not

show enhanced hygroscopicity and that the chemical composition of the dust remains rather unchanged. Therefore, we do not assume any effects caused by the high relative humidity within the CMBL. Furthermore the high relative humidity within the boundary layer indicates that we do not have to consider any dry marine aerosols within the boundary layer (Murayama et al., 1999; Sakai et al., 2010).

To determine the dust volume and the dust volume fraction the derived dust extinction coefficient has to be converted to dust volume. For this conversion we used a conversion factor derived from AERONET sunphotometer measurements. To confirm this conversion factor we compare it to the modeled conversion factor using the method described by Gasteiger et al. (2011). Assuming a reference ensemble to calculate the volume and extinction coefficient, the conversion factor is then the ratio between the derived volume and extinction. Gasteiger et al. (2011) used an ensemble which was consistent with lidar

measurements of Saharan aerosol in Morocco during SAMUM-1. As the airborne in-situ measurements show no significant changes of the properties of long-range transported Saharan dust (Weinzierl et al., 2016) we assume that this reference ensemble is still valid for long-range transported Saharan dust. With the method described by Gasteiger et al. (2011) we found an extinction to volume conversion factor of $0.68 \cdot 10^{-6}$ m for a wavelength of 532nm. This value is very similar to the one derived from sunphotometer measurements, confirming its validity for this study.

To further confirm our method and the derived results we compared the lidar derived dust mass concentration with coordinated in-situ measurements. For uncorrected in-situ dust mass concentrations we find that, aside from a negative bias of the in-situ measurement of about 3 $\mu g/m^3$, both methods agree well up to a dust mass concentration of about 40 $\mu g/m^3$. A correction factor taking into account the underrepresented size range of the dust size distribution from the in-situ measurements could compensate the bias in the in-situ measurements. The differences at larger values of the dust mass concentration can partly be

explained by the underestimation of large dust mass concentration from PM10 measurements. However, the differences for very large values of around 100 $\mu g/m^3$ can not fully be explained by either the underestimation of the PM10 measurements or the correction due to the uncovered size range. Those high values were measured in conditions with low wind speed compared to the other days during this study. Thus it may be possible that effects during transport across Barbados, e.g. from turbulence over the island or island heat effects may be of importance for measurements at the downwind side of the island.

**5   Conclusions**

Different measurements and methods in different height ranges are often difficult to link and to compare directly. Especially different aerosol mixtures in different height ranges pose difficulties in the comparison of measurements. Lidar measurements are a valuable tool as they provide height resolved information over the entire atmospheric column. Furthermore, with the method applied in this study it is not only possible to characterize the different layers within the atmospheric column, but also to

derive the contributions of the different aerosol types to aerosol mixtures. This is of importance to link e.g. measurements of one single aerosol type to measurements of the whole aerosol mixture, measurements of different height ranges, or measurements of single height ranges to measurements of the whole column. The measurements presented in this work can, in particular, be used to study the downward mixing of Saharan dust after long-range transport across the Atlantic Ocean or to study dust removal

processes. The results derived here can also serve to validate dust transport models as they separate the dust contribution from the contribution of other aerosol types. Furthermore the results provide insight into dust removal processes as they provide a detailed characterization of the conditions within the CMBL during the SALTRACE measurement period.

*Acknowledgements.* The SALTRACE campaign was mainly funded by the Helmholtz Association, the Deutsches Zentrum für Luft- und Raumfahrt (DLR), the Ludwig-Maximilians-Universität München (LMU) and the Institut für Troposphärenforschung (TROPOS). This work project was partly founded by a DLR VO-R young investigator group, by the Deutsche Forschungsgemeinschaft (DFG) in the SPP (no. 1294/2) "Atmosphären- und Erdsystemforschung mit dem Forschungsflugzeug HALO" under contract no. KI1567/1-1, and the LMU Munichs Institutional Strategy LMUexcellent within the framework of the German Excellence Initiative. The lidar and sun-photometer measurements were performed at the site of the Caribbean Institute for Meteorology and Hydrology (CIMH). We thank CIMH for providing us with this measurement environment.

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
