# Peer review of "Saharan dust contribution to the Caribbean summertime boundary layer - A lidar study during SALTRACE"

_Atmospheric Chemistry and Physics, 2016_

## Referee Comment (RC1) · Anonymous Referee #1 · 9 May 2016

The paper by *Groß et al.* investigates the contribution of Saharan dust to the boundary layer over Barbados as observed during SALTRACE. The paper is of interest to the scientific community but major revisions are necessary before further consideration for publication in ACP.

**Major points:**

- A description of the used instrumentation is completely missing in the text. Section 2 should be revised to Instruments and Methods. There should be at least a table that provides an overview of the used instrumentation. The authors only mention auxiliary measurements with sun photometer and in situ measurements

when they are already discussing results in Section 3.

- It is not acceptable to use papers in preparation as references. Nothing is known about the status of these papers

- The authors should consider restructuring the paper. It seems more straight-forward to first discuss the measured optical properties and later describe the subsequently retrieved parameters. This means that all optical properties should be addressed before Figure 4 is discussed.

- Greater care is necessary with respect to the investigated height range. The authors loosely vary between the terms convective marine boundary layer, convective boundary layer and just boundary layer. Are these meant to be the same things? Later the also discuss the transition layer and the Saharan air layer. It might be worthwhile to properly define all these layers in the example provided in Figure 2.

- Please make sure that the same tense is used throughout the paper.

- Statements of good and very good agreement need to be quantified.

**Minor points:**

- Check the co-authors' affiliations. I believe it's Leibniz Institute.

- p1,l13: 80% seems like a normal value for RH in marine environment.

- p1,l20: Are the measurements just used to support modelling efforts or rather to validate them?

- p2,l11: Please elaborate on the point of efficient downward mixing.

- p3,Section 2.2: More background is needed on how the conversion factors have been obtained. Did you apply any constraints for retrieving marine conversion factors from AERONET measurements at Barbados? Why are the factors almost identical for marine aerosol and mineral dust?

- p4,l4: Does this mean that you use the gradient method to find the top height of the CMBL? Do you use the first gradient or the strongest gradient? please provide more information.

- p4,Section 3.2: More details are needed regarding the analysis of the lidar measurements. You could provide those in an Instruments section: What is the averaging time of the lidar measurements? Were the lidar measurements performed during day or night? How did you analyze the data? Which lidar ratio has been used to derive the backscatter profiles?

- p5,l12: Could the differences in lidar and sounding be the result of the two hours time delay between the two?

- p6,Figure 3: What is the general time difference between the lidar measurements and the soundings?

- p7,l3: Please elaborate what is meant with intensive lidar quantities for the unfamiliar reader.

- p7,l12: I don't believe that this paper is the best reference on sea spray production.

- p8,Figure 5: Add mean/median/sd to the figure. Improve the scale in lidar ratio, i.e. 0 to 50 sr.

- p10,l10: More details are needed for the in situ measurements used in the closure study. Which instruments are involved? How have those measurements been

transformed to mass concentration? What is meant with "match in time"? Such criteria need to be provided in the paper.

---

## Referee Comment (RC2) · Anonymous Referee #2 · 11 May 2016

The authors describe a case study of Saharan dust observed over the Caribbean with a dual wavelength lidar. In their paper, the authors describe time series of LIDAR (CIMEL and POLIS) measurements to highlight a study case (10 July 2013) and vertical profiles of this study case. They also provide a closure study based on the comparison of LIDAR retrieved parameters and in-situ measurements. This manuscript is of interest for the scientific community but need major revisions before submission to ACP.

MAJOR COMMENTS :

1. The scientific objectives of the study are limited to "provide detailed BL character­ization as part of the vertical aerosol structure over Barbados during SALTRACE" as it play a significant role in the synergy between ground based, airborne and column

integrated measurements. Could you state clearly how your results will help to link all these measurements? Could you also state if and how those results may be applied to different measurement campaign?

2. This paper is referring to LIDAR (POLIS and CIMEL), in-situ and radio-soundings measurements. There is no description of the used instruments, which is mandatory. Also every all the algorithms to correct the data, if existing, must be described in one specific section.

3. Figure 4 : From the dust mass concentration shown in this figure, one can see that the variability is not important from day to day. The dust mass concentration is on average 40ug/cm3. Two outliers can be distinguished at 70and 100ug/cm3. That would have been really interesting to show the lidar profile for these two cases when dust are obviously mixed with sea salt.

Looking closely to the values for the study case (10/07/2013) the values are always below 40ug/cm3 and increasing throughout the day. Now from the profiles shown in Figure 2 the average mass concentration of dust within the CMBL is about 110ug/cm-3. This strong difference makes questionable the quality of the data used in Figure 2 or in Figure 4.

MINOR COMMENTS :

1. Could you provide a map to show the location where SALTRACE took place ?

2. Although, Denjean et al 2015 found (based on model results) that optical properties of one dust plume particles were not modified during their transport over Atlantic, many studies have shown differences in the dust size distributions, in the dust morphology, and also on the dust optical properties including dust polarization (Bréon et al. 2013). Why you are stating that here ? Is it related to your choice of a mean depolarization ratio of 0.30 ? If yes then you should lead the reader into it cause I don't see the point here. Also, Burton et al. (2015), using HSRL measurements, highlight a dust

particulate depolarization ratio of 0.32 over the Caribbean islands. You should spend more energy on why you choose 0.30.

3. P3, L8 Remains not remaines

4. Dust volume and mass conversion: You use a conversion factor of 0.65.10-6 m. According to previous studies large aerosols do not reach the Caribbean coast (Maring et al. 2003). So this factor should not remain the same over the dust source and after few days of transport. Could you precise if the Gasteiger study was referring to fresh or aged dust ? Moreover it appears difficult to compare a factor obtained from measurements over the source (SAMUM) to a factor obtained after transport (SALTRACE). How did you calculate this factor ? It seems that you use the aerosol volume from AERONET measurements (integrated over the column) and an extinction coefficient from the LIDAR (at which altitude ?). What are the errors associated with this coefficient (AERONET volume errors + Lidar ratio errors + density errors )? Did you perform a closure with the AOD from AERONET and the AOD from your LIDAR data ?

This factor is depending on the altitude, right ? Bigger SS at the surface, dust mixed with SS and pure dust over those layers.

5. You choose to use a density of 2.5g/cm3 assuming that dust are mixed with sulphate. Earlier you state that dust chemistry was not changing during the transport. If there is any sulphate on a dust particle, even a little bit, then the dust becomes hygroscopic (Roberts et al., 2001) and the optical properties are not the same than pure dust. You need to clarify this point

6. You need to show the data that provide you enough information to chose 0.65 .10-6 m for dust and 0.66 .10-6 m for sea salt particles.

7. Could you provide the extinction profile retrieved for the 10 July 2013 study case ?

8. P6 L4 You say that dust particles contribute to 100% of the total aerosol volume so why is there no pure dust in Table 2 ?

9. From what I understand, you used the wind speed to say that SS can be generated at the surface, the wind directions to say that wind are mainly coming from East / North-East, and the relative humidity is always larger than 60% and in average 80%. Does that need to be plotted ?

10. What can we learn based on your CMBL height retrievals ? You say that some cases are not well retrieved by the LIDAR and you say why. Is there a solution to avoid those mistakes with this kind of LIDAR ?

11. Figure 5 : Could you change the scale of the Lidar ratio plot ? The values are between 15-35 and your scale is between 0-100. What are the green and blue dots represent?

12. Section 3.6 : What kind of in-situ measurement did you use ?

- The correction you apply to these unknown in situ data is based on OPAC desert mixture and assuming 10um particles reach the barbadoes. What is this correction about? You cannot use a correction factor that please you without explaining the reader what you exactly did. - You are only looking to data that 'match in time'. What does that mean ? Is it a window of an hour, 10 minutes? - This closure doesn't conveniently take into account the larger values of dust concentration. In the figure caption remove the second 'dust'

13. The summary is not giving any conclusions or any clue to better improve the relation between ground base and remote sensing measurements. You should be careful in the summary and say that you were able with this case to link ground based measurements to remote sensing measurement but that for other cases it might not be as easy to achieve. You also need to tell the reader why you were able to do it (mixing condition with the BL, Just two type of particles etc. . . )

---

## Referee Comment (RC3) · Anonymous Referee #3 · 15 May 2016

**General comments**

The paper presents a case study related to SALTRACE campaign addressed to the characterization of the boundary layer with the presence of a mix of aerosol (dust and maritime) in the Caribbean area during a Saharan dust transport. It deals with a very interesting topic for the scientific community involved in atmospheric research because it provides information that can be merged with other results coming from other papers produced for the same campaign, obtaining a large and exhaustive overview and interpretation of the atmospheric observations in a particular site and in several kinds of conditions. This gives the paper a value even if it is not particularly original.

The paper seems to be written with no sufficient detail in the discussions and justifications. Improvement and more care should be requested in English language, being present several English grammar typos errors.

**Specific comments**

*Some general considerations*

1) In order to give the paper more completeness and to allow a better understanding of the observations, the authors should link (and cite) a previous paper: *"S. Groß, V. Freudenthaler, K. Schepanski, C. Toledano, A. Schäfler, A. Ansmann, and B. Weinzierl, Optical properties of long-range transported Saharan dust over Barbados as measured by dual-wavelength depolarization Raman lidar measurements, Atmos. Chem. Phys., 15, 11067–11080, 2015, www.atmos-chem-phys.net/15/11067/2015/, doi:10.5194/acp-15-11067-2015"*, where most of the authors are the same, the lidar system is the same and also the measurement period is the same. In the present paper the authors address the study to a different day, characterized by a dominance of marine aerosols.

2) There are several references to papers in preparation for the same special issue. In my opinion, this is possible if some aspects presented will be furtherly analyzed and discussed in those, but this is strange if the results of those papers are used (e.g. Groß et al 2015, Haarig et al., Marinou et al.) before the corresponding peer review processes. In principle, the results or conclusions of those papers could also be rejected. This paper should have its own self-consistence and therefore the results of those other papers should be introduced in a different way, otherwise the paper should be accepted after the others will be accepted for publication.

*In detail*

Page 2, Lines 11-12: The authors write: "This strong increase at the top of the cloud-topped or cloud-less CMBL is to our opinion a clearly sign for an efficient..." The conclusion should be better discussed by the authors.

Page 3, Line 6: The authors justify the assumption about the two component mixture of marine aerosols and mineral dust with "coordinated in-situ measurements". Which kind of measurements?

Section 2.1. The authors assume several values for linear depolarization ratio for dust and marine aerosols, lidar ratio. Did they try to have support from direct measurements to these assumptions? For example, in my opinion, with reference to the paper I cited before (Point 1), why in this paper the authors do not use a similar optical characterization?

Section 2.2: It is not clear to me how, from the reference ensemble of Gesteiger et al (2011) at 532 nm, the value 0.68x10-6m is obtained. But, in general, this is applicable also to the several values of v/alpha. The assumptions are introduced in a very fast way, without justifications. I think, a discussion, even if minimum, should be given to give the paper a self-consistence.

Page 4, Line 15: Did the authors tried a comparison using Raman measurements? According to the paper I cited at the beginning (Point 1), POLIS is also equipped with Raman channels. How the backscatter in fig 2a are calculated? Why Raman measurements have been not used to characterize the layers like in the previous paper (see Point 1)?

Page 9, Table 1: It is not clear to me the case 24 June – 10 July. What does this indication mean: dust and marine (marine dominated), but without marine cases.

Page 10, Line 2: Which is the distance between the measurement site and the Ragged Point? Is this comparison significant?

Page 10, Line 8: Which is the meaning of the factor 1.25? How is it obtained?

Page 10, Lines 9 and 10: What does it mean "we assume an uncertainty of...". How this estimation has been obtained?

Page 10: The comment to the results of Fig. 6 is really very short. In general, these should be better discussed.

Page 11, Summary: I image that the conclusions are referred to the 10 July case study. The authors does not report this. Moreover, I do not see correspondence between the values reported for PLDR in the Summary and those reported in table 1. Again, in the last line, which is the distance from the eastern part of the island? In general, the summary should be more clear and should give the idea of the importance of the reached goals.

**Technical corrections**
Page 1, Line 5: change "information of the CMBL" into "information on the CMBL"

Page 2, Line 21: change "information of the boundary layer" into "information on the boundary layer"

Page 2, Line 24: change "ground-base" into "ground-based"

[Figure]

Page 2, Line 26: change "located at the area" into "located in the area".

Page 3 Lines 6-8: Specify that the content of the sentence has been demonstrated when aerosols are transported across the Atlantic in summertime, otherwise it seems valid in general.

Page 7, Lines 3 and 5: change "on top the CMBL" into "on top of the CMBL".

Page 7, Line 3: change "found, that" into "found that". Remove the comma.

Page 8, Line 9: the authors write "AOD >= 0.4 nm". They missed the wavelength between "0.4" and "nm"

---

## Author Comment (AC1) · 22 Jul 2016

*We thank this Reviewer for his careful reading of the manuscript and for his suggestions to help us improve the paper.*

*The answers are given in a direct response (bold, italic).*

The paper by Groß et al. investigates the contribution of Saharan dust to the boundary layer over Barbados as observed during SALTRACE. The paper is of interest to the scientific community but major revisions are necessary before further consideration for publication in ACP.

Major points:

A description of the used instrumentation is completely missing in the text. Section 2 should be revised to Instruments and Methods. There should be at least a table that provides an overview of the used instrumentation. The authors only mention auxiliary measurements with sun photometer and in situ measurements when they are already discussing results in Section 3.

*We added a description of all used instrumentation and methods in Section 2.*

It is not acceptable to use papers in preparation as references. Nothing is known about the status of these papers

*We removed the papers in preparation as references but we kept the announcement that papers dealing with the same topic are in preparation.*

The authors should consider restructuring the paper. It seems more straight-forward to first discuss the measured optical properties and later describe the subsequently retrieved parameters. This means that all optical properties should be addressed before Figure 4 is discussed.

*We agree with this reviewer that a restructuring of the paper would be more straight-forward and followed his suggestion to first discuss the optical properties and then the subsequently retrieved parameters.*

Greater care is necessary with respect to the investigated height range. The authors loosely vary between the terms convective marine boundary layer, convective boundary layer and just boundary layer. Are these meant to be the same things? Later they also discuss the transition layer and the Saharan air layer. It might be worthwhile to properly define all these layers in the example provided in Figure 2.

*We are more consistent now.*

Please make sure that the same tense is used throughout the paper.

*We revised the paper to check the tense.*

Statements of good and very good agreement need to be quantified.

*We replaced these statements by more precise statements.*

Minor points:

• Check the co-authors' affiliations. I believe it's Leibniz Institute.

*We changed that.*

• p1,l13: 80% seems like a normal value for RH in marine environment.

*Indeed, 80% is a normal value for RH in marine environment. We mentioned that value as it justifies the use of optical properties for moist sea salt. We removed this statement in the abstract but mention this in the text.*

• p1,l20: Are the measurements just used to support modelling efforts or rather to validate them?

*Indeed, the measurements are also used to validate modelling efforts. We changed the text accordingly.*

• p2,l11: Please elaborate on the point of efficient downward mixing.

*We believe that the high values in the particle linear depolarization ratio in the layer below the well-defined Saharan Air Layer is already an indication that dust removal processes started to mix the dust out of the SAL down to the ground. We added this to the text.*

p3,Section 2.2: More background is needed on how the conversion factors have been obtained. Did you apply any constraints for retrieving marine conversion factors from AERONET measurements at Barbados? Why are the factors almost identical for marine aerosol and mineral dust?

*We provide now more information how the conversion factors are calculated.*

*To the question of the almost identical values: For large particles the aerosol extinction coefficient is mainly dominated by the size of the particles. As both, marine aerosols and mineral dust, are large particle types in the same size range also their conversion factor from extinction to volume should be almost the same.*

• p4,l4: Does this mean that you use the gradient method to find the top height of the CMBL? Do you use the first gradient or the strongest gradient? Please provide more information.

*The gradient method was applied to derive the top height of the CMBL which was defined as height range of the strongest gradient. Furthermore we used the change of the change of the intensive optical properties to strengthen our result. We changed the text to be more precise there.*

• p4,Section 3.2: More details are needed regarding the analysis of the lidar measurements. You could provide those in an Instruments section: What is the averaging time of the lidar measurements? Were the lidar measurements performed during day or night? How did you analyze the data? Which lidar ratio has been used to derive the backscatter profiles?

*We added a subsection in the 'Instrumentation and Method' section to describe the lidar system, analyzes of the data and specific information on the used data.*

• p5,l12: Could the differences in lidar and sounding be the result of the two hours time delay between the two?

*We do not believe that the missing capping inversion in the radiosonde at the CMBL top height derived from the gradient method is due to a two hour shift between both measurements. We*

*rather think that there was no strong capping inversion on that day. This would also explain the little difference in the intensive lidar quantities below the CMBL top height and above.*

• p6,Figure 3: What is the general time difference between the lidar measurements and the soundings?

*Typically the soundings were launched during the lidar measurement sessions, thus no or only little time difference is found in general. We added a subsection in the section 'Instruments and Method' which includes this information.*

• p7,l3: Please elaborate what is meant with intensive lidar quantities for the unfamiliar reader.

*Intensive lidar quantities are only dependent on the type of the observed aerosol or aerosol mixture. The intensive lidar quantities are not dependent on the amount of aerosols. We added a description in the text.*

• p7,l12: I don't believe that this paper is the best reference on sea spray production.

*We use this paper here as the authors conducted an empirical study of the relation between sea salt production and wind speed. We use their thresholds in this study.*

• p8,Figure 5: Add mean/median/sd to the figure. Improve the scale in lidar ratio, i.e. 0 to 50 sr.

*We show now mean, stdev, median and error of the mean in this figure and improved the lidar ratio scale from 0 to 50 sr.*

• p10,l10: More details are needed for the in situ measurements used in the closure study. Which instruments are involved? How have those measurements been transformed to mass concentration? What is meant with "match in time"? Such criteria need to be provided in the paper.

*We added a subsection in the 'Instrumentation and Method' section to provide information on the in-situ measurements and their analysis.*

---

## Author Comment (AC2) · 22 Jul 2016

*We thank this Reviewer for his careful reading of the manuscript and for his suggestions to help us improve the paper.*

*The answers are given in a direct response (bold, italic).*

The authors describe a case study of Saharan dust observed over the Caribbean with a dual wavelength lidar. In their paper, the authors describe time series of LIDAR (CIMEL and POLIS) measurements to highlight a study case (10 July 2013) and vertical profiles of this study case. They also provide a closure study based on the comparison of LIDAR retrieved parameters and in-situ measurements. This manuscript is of interest for the scientific community but need major revisions before submission to ACP.

MAJOR COMMENTS :

The scientific objectives of the study are limited to "provide detailed BL characterization as part of the vertical aerosol structure over Barbados during SALTRACE" as it play a significant role in the synergy between ground based, airborne and column integrated measurements. Could you state clearly how your results will help to link all these measurements? Could you also state if and how those results may be applied to different measurement campaign?

*We changed the text to provide more infromation.*

This paper is referring to LIDAR (POLIS and CIMEL), in-situ and radio-soundings measurements. There is no description of the used instruments, which is mandatory. Also every all the algorithms to correct the data, if existing, must be described in one specific section.

*We included this information in the 'Instrumentation and Method' section.*

Figure 4 : From the dust mass concentration shown in this figure, one can see that the variability is not important from day to day. The dust mass concentration is on average 40ug/cm3. Two outliers can be distinguished at 70and 100ug/cm3. That would have been really interesting to show the lidar profile for these two cases when dust are obviously mixed with sea salt.

*We decided to not show another case study as we already showed a case with large dust contribution in the boundary layer. We included now the values from the case study in Figure 4 (now Figure 5).*

Looking closely to the values for the study case (10/07/2013) the values are always below 40ug/cm3 and increasing throughout the day. Now from the profiles shown in Figure 2 the average mass concentration of dust within the CMBL is about 110ug/cm-3. This strong difference makes questionable the quality of the data used in Figure 2 or in Figure 4.

*The reviewer is right that the value shown in Figure 2 is missing in Figure 4 (now Figure 5) as we used only synchronized lidar and radiosonde measurements for consistency of the analysis shown in Figure 3 and (now) Figure 5.*

MINOR COMMENTS :

Could you provide a map to show the location where SALTRACE took place ?

*Instead of showing a map of the SALTRACE location we referenced the SALTRACE overview paper in BAMS providing all necessary information about the SALTRACE campaign.*

Although, Denjean et al 2015 found (based on model results) that optical properties of one dust plume particles were not modified during their transport over Atlantic, many studies have shown differences in the dust size distributions, in the dust morphology, and also on the dust optical properties including dust polarization (Bréon et al. 2013). Why you are stating that here ? Is it related to your choice of a mean depolarization ratio of 0.30 ? If yes then you should lead the reader into it cause I don't see the point here.  Also, Burton et al.  (2015), using HSRL measurements, highlight a dust particulate depolarization ratio of 0.32 over the Caribbean islands. You should spend more energy on why you choose 0.30.

*We refered to the work of Denjean et al 2015 to justify that we do not consider changes in the particle depolarization ratio due to the high relative humidity. We thus took a mean value for the linear particle depolarization ratio of dust of 0.3 found during former studies (e.g.Freudenthaler et al., 2009; Liu et al., 2008, Groß et al., 2011) which is in good agreement with the values we found during the same campaign within the Saharan air layer over Barbados from our measurements (Groß et al., 2015). We tried to this more clear in the text.*

P3, L8 Remains not remains

*We changed that.*

Dust volume and mass conversion: You use a conversion factor of 0.65.10-6 m. According to previous studies large aerosols do not reach the Caribbean coast (Maring et al. 2003). So this factor should not remain the same over the dust source and after few days of transport. Could you precise if the Gasteiger study was referring to fresh or aged dust ? Moreover it appears difficult to compare a factor obtained from measurements over the source (SAMUM) to a factor obtained after transport (SALTRACE). How did you calculate this factor ? It seems that you use the aerosol volume from AERONET measurements (integrated over the column) and an extinction coefficient from the LIDAR (at which altitude ?). What are the errors associated with this coefficient (AERONET volume errors + Lidar ratio errors + density errors )? Did you perform a closure with the AOD from AERONET and the AOD from your LIDAR data ?

*From the measurements of size distribution we do not confirm that no large particles reach the Caribbean coast (see Weinzierl et al., 2016, van der Does et al., 2016). Analyses of the conversion factors from sunphotometer measurements confirm that the value does not change significantly during transport and that the conversion factor derived with the method described in Gasteiger et al. is still valid for long-range transported Saharan dust although the reference ensemble of Gasteiger et al. is referring to dust near the source region (measurements performed in Morocco). The conversion factor derived with the method described by Gasteiger et al. is the ratio of volume to extinction. The value from sunphotometer is also derived by first calculating the volume, the extinction, and the ratio of both using the aerosol mixture retrieved by the AERONET inversion algorithm for the calculation. We provide now an improved description of the method. For detailed information about the methods we included the references to the corresponding publications.*

This factor is depending on the altitude, right ? Bigger SS at the surface, dust mixed with SS and pure dust over those layers.

*We believe that in the convective marine boundary layer, which is characterized by mixing processes, we do not have to take any height dependency of the particle properties or conversion factors into account, especially not in the height ranges we are able to observe with our lidar system. Maybe this is different in the lowermost meters above ground, but we miss these height ranges with our observations.*

You choose to use a density of 2.5g/cm3 assuming that dust are mixed with sulphate. Earlier you state that dust chemistry was not changing during the transport. If there is any sulphate on a dust particle, even a little bit, then the dust becomes hygroscopic (Roberts et al., 2001) and the optical properties are not the same than pure dust. You need to clarify this point

*Measurements of the chemical properties of dust near the source region (Kaaden et al., 2009) and over Barbados (Kandler et al., in preparation) show that there is always a portion of sulfate externally mixed (Weinzierl et al., 2009) in the Saharan air layer. Thus, the statement that we do not see changes in the chemistry also includes that we do not see changes in this external mixture and that the dust particles are not coated with the sulfate at the end of the long-range transport across the Atlantic Ocean.*

You need to show the data that provide you enough information to chose 0.65 .10-6 m for dust and 0.66 .10-6 m for sea salt particles.

*We decided to not show all the measurements of the measurement period from 2007 to 2015. These measurements are freely available on the AERONET webpage and can be reviewed there.*

Could you provide the extinction profile retrieved for the 10 July 2013 study case ?

*We included the extinction coefficient profile in the case study.*

P6 L4 You say that dust particles contribute to 100% of the total aerosol volume so why is there no pure dust in Table 2 ?

*Table 2 only includes measurements within the CMBL while the dust contribution of 100% refers to height ranges above 1.6 km in the shown case study.*

From what I understand, you used the wind speed to say that SS can be generated at the surface, the wind directions to say that wind are mainly coming from East / North- East, and the relative humidity is always larger than 60% and in average 80%. Does that need to be plotted ?

*This information is already shown in Figure 3.*

What can we learn based on your CMBL height retrievals ? You say that some cases are not well retrieved by the LIDAR and you say why. Is there a solution to avoid those mistakes with this kind of LIDAR ?

*We do not think that you can avoid these mistakes. However what we show and also state in the text is that if the gradient is so small that  it would lead to this kind of mistake, then the intense optical properties do not change significantly between the lower layer and the layer above.*

Figure 5 : Could you change the scale of the Lidar ratio plot ? The values are between 15-35 and your scale is between 0-100. What are the green and blue dots represent?

*We changed the scale for the lidar ratio to values from 0 to 50 sr. We clearly indicate the meaning of the different symbols in the figure caption.*

Section 3.6 : What kind of in-situ measurement did you use ?

*The description of the in-situ measurements is now added in Section 2.*

- The correction you apply to these unknown in situ data is based on OPAC desert mixture and assuming 10um particles reach the barbadoes. What is this correction about? You cannot use a correction factor that please you without explaining the reader what you exactly did. - You are only looking to data that 'match in time'. What does that mean ? Is it a window of an hour, 10 minutes? - This closure doesn't conveniently take into account the larger values of dust concentration. In the figure caption remove the second 'dust'

*For the derivation of this correction factor we use the OPAC desert mixture and calculate the aerosol volume of this mixture for upper cut-off radii of 5 and 10 micrometer. We calculate the ratio between both volumes, assuming that a cut-off radius of 10 micrometer is valid for dust reaching Barbados and a cut-off radius of 5 micrometer is valid for the instrument. This factor is about 1.25 and is applied to the PM10 measurements to calculate the ambient dust volume. However, as the uncertainty about the size distribution of dust after long-range transport is large, we consider an uncertainty of +-0.25 which also covers the case that no aerosol with r > 5 micrometer reaches Barbados.*

The summary is not giving any conclusions or any clue to better improve the relation between ground base and remote sensing measurements. You should be careful in the summary and say that you were able with this case to link ground based measurements to remote sensing measurement but that for other cases it might not be as easy to achieve. You also need to tell the reader why you were able to do it (mixing condition with the BL, Just two type of particles etc.

*We reworked the Summary.*

---

## Author Comment (AC3) · 22 Jul 2016

*We thank this Reviewer for his careful reading of the manuscript and for his suggestions to help us improve the paper.*

*The answers are given in a direct response (bold, italic).*

General comments

The paper presents a case study related to SALTRACE campaign addressed to the characterization of the boundary layer with the presence of a mix of aerosol (dust and maritime) in the Caribbean area during a Saharan dust transport. It deals with a very interesting topic for the scientific community involved in atmospheric research because it provides information that can be merged with other results coming from other papers produced for the same campaign, obtaining a large and exhaustive overview and interpretation of the atmospheric observations in a particular site and in several kinds of conditions. This gives the paper a value even if it is not particularly original.

The paper seems to be written with no sufficient detail in the discussions and justifications. Improvement and more care should be requested in English language, being present several English grammar typos errors.

*We changed the manuscript considering the suggestions of this Reviewer. We modified the Summary and included a Section to discuss the results. We also checked on grammar typos errors.*

Specific comments

Some general considerations

In order to give the paper more completeness and to allow a better understanding of the observations, the authors should link (and cite) a previous paper:  "S. Groß, V. Freudenthaler, K. Schepanski, C. Toledano, A. Schäfler, A. Ansmann, and B. Weinzierl, Optical properties of long-range transported Saharan dust over Barbados as measured by dual-wavelength depolarization Raman lidar measurements, Atmos. Chem. Phys., 15, 11067–11080, 2015, www.atmos-che-phys.net/15/11067/2015/, doi:10.5194/acp-15-11067-2015", where most of the authors are the same, the lidar system is the same and also the measurement period is the same. In the present paper the authors address the study to a different day, characterized by a dominance of marine aerosols.

*As suggested by this reviewer we linked and cited the previous paper in this work.*

There are several references to papers in preparation for the same special issue. In my opinion, this is possible if some aspects presented will be furtherly analyzed and discussed in those, but this is strange if the results of those papers are used (e.g. Groß et al 2015, Haarig et al., Marinou et al.) before the corresponding peer review processes. In principle, the results or conclusions of those papers could also be rejected. This paper should have its own self-consistence and therefore the results of those other papers should be introduced in a different way, otherwise the paper should be accepted after the others will be accepted for publication.

*We agree that the paper has to be self-consistent and thus we replaced the papers in preparation by peer-reviewed published papers*

*where a link to former or other work is needed. However we kept the announcement that papers dealing with the same topic are in preparation for this special issue.*

In detail

Page 2, Lines 11-12: The authors write: "This strong increase at the top of the cloud-topped or cloud-less CMBL is to our opinion a clearly sign for an efficient..." The conclusion should be better discussed by the authors.

*We believe that the high values in the particle linear depolarization ratio in the layer below the well-defined Saharan Air Layer is already an indication that dust removal processes started to mix the dust out of the SAL down to the ground. We added this to the text.*

Page 3, Line 6: The authors justify the assumption about the two component mixture of marine aerosols and mineral dust with "coordinated in-situ measurements". Which kind of measurements?

*For this assumption we used airborne in-situ measurements of microphysical and chemical composition of the observed aerosols. We added this in the text.*

Section 2.1. The authors assume several values for linear depolarization ratio for dust and marine aerosols, lidar ratio. Did they try to have support from direct measurements to these assumptions? For example, in my opinion, with reference to the paper I cited before (Point 1), why in this paper the authors do not use a similar optical characterization?

*This reviewer is right that we should link the previous paper to this work to justify our assumptions of the used values for the linear*

*depolarization ratio of dust and marine aerosols. We measured these values during the SALTRACE campaign and the values are described in detail in Groß et al., 2015. We now refer to this optical haracterization of the different aerosol types to make clear that the values used for the type separation are based on direct measurements at Barbados.*

Section 2.2: It is not clear to me how, from the reference ensemble of Gesteiger et al (2011) at 532 nm, the value 0.68x10-6m is obtained. But, in general, this is applicable also to the several values of v/alpha. The assumptions are introduced in a very fast way, without justifications. I think, a discussion, even if minimum, should be given to give the paper a self-consistence.

*The v/alpha value is calculated for the reference ensemble by first calculating the volume of the mixture and then the extinction coefficient of the mixture as described by Gasteiger et al.*

Page 4, Line 15: Did the authors tried a comparison using Raman measurements? According to the paper I cited at the beginning (Point 1), POLIS is also equipped with Raman channels. How the backscatter in fig 2a are calculated? Why Raman measurements have been not used to characterize the layers like in the previous paper (see Point 1)?

*POLIS is indeed equipped with Raman channels and we used these measurements for the optical characterization of the different aerosols and aerosols layers as was shown in the previous paper mentioned by this reviewer (Point 1). We also characterized the intensive optical lidar properties in the boundary layer as shown in Fig. 5 (now Fig. 4) and Table 1. The shown case study was performed during daytime where no Raman measurements were performed. As the aerosol type separation is based on depolarization measurements, the backscatter*

*value and depolarization ratio measurements are the most important values. We derived them with the Fernald/Klett algorithm as is now described in the 'Instrumentation and Method' section. We chose this case study as the measurements were performed during aircraft overflights over the ground-based station and the date was chosen as one of the 'golden cases.' Thus this case study might be useful for further analyzes. We tried to better link to the previous study.*

Page 9, Table 1: It is not clear to me the case 24 June – 10 July. What does this indication mean: dust and marine (marine dominated), but without marine cases.

*'dust and marine (marine dominated)' refers to a mixture of dust and marine aerosols which optical properties dominated by the marine aerosols in this mixture.*

Page 10, Line 2: Which is the distance between the measurement site and the Ragged Point? Is this comparison significant?

*The distance between Ragged Point and the lidar measurement site is about 40 km. To check the significance of comparisons between both sites we looked on aircraft in-situ and sunphotometer measurements of total AOD and Angström Exponent.*

Page 10, Line 8: Which is the meaning of the factor 1.25? How is it obtained?

*For the derivation of this correction factor we use the OPAC desert mixture and calculate the aerosol volume of this mixture for upper cut-off radii of 5 and 10 micrometer. We calculate the ratio between both volumes, assuming that a cut-off radius of 10 micrometer is valid for dust reaching Barbados and a cut-off radius of 5 micrometer is valid*

*for the instrument. This factor is about 1.25 and is applied to the PM10 measurements to calculate the ambient dust volume. However, as the uncertainty about the size distribution of dust after long-range transport is large, we consider an uncertainty of +-0.25 which also covers the case that no aerosol with r > 5 micrometer reaches Barbados.*

Page 10, Lines 9 and 10: What does it mean "we assume an uncertainty of. . .". How this estimation has been obtained?

*See previous comment.*

Page 10: The comment to the results of Fig. 6 is really very short. In general, these should be better discussed.

*We extended the discussion.*

Page 11, Summary: I image that the conclusions are referred to the 10 July case study. The authors does not report this. Moreover, I do not see correspondence between the values reported for PLDR in the Summary and those reported in table 1. Again, in the last line, which is the distance from the eastern part of the island? In general, the summary should be more clear and should give the idea of the importance of the reached goals.

*We completely modified the summary to make it more clear and to give and idea of the reached goals.*

Technical corrections

Page 1, Line 5: change "information of the CMBL" into "information on the CMBL"

*We changed that.*

Page 2, Line 21: change "information of the boundary layer" into "information on the boundary layer"

*We changed that.*

Page 2, Line 24: change "ground-base" into "ground-based"

*We changed that.*

Page 2, Line 26: change "located at the area" into "located in the area".

*We corrected that.*

Page 3 Lines 6-8: Specify that the content of the sentence has been demonstrated when aerosols are transported across the Atlantic in summertime, otherwise it seems valid in general.

*We tried to limit the validity of this statement by mentioning that it is valid for this study. However we agree with this Reviewer that a wrong assumption of general validity has to be avoided. Therefore we modified the text to make clear that this two-type assumption is only valid during dust long-range transport across the Atlantic Ocean as it was found for this study.*

Page 7, Lines 3 and 5: change "on top the CMBL" into "on top of the CMBL". Page 7, Line 3: change "found, that" into "found that". Remove the comma.

*We corrected that.*

Page 8, Line 9: the authors write "AOD >= 0.4 nm". They missed the wavelength between "0.4" and "nm"

*We corrected that.*